# The Influence of the Degree of Forest Management on Methylmercury and the Composition of Microbial Communities in the Sediments of Boreal Drainage Ditches

**DOI:** 10.3390/microorganisms10101981

**Published:** 2022-10-06

**Authors:** Krišs Bitenieks, Arta Bārdule, Karin Eklöf, Mikk Espenberg, Dainis Edgars Ruņģis, Zane Kļaviņa, Ivars Kļaviņš, Haiyan Hu, Zane Lībiete

**Affiliations:** 1Latvian State Forest Research Institute ‘Silava’ (LSFRI Silava), Rigas Str. 111, LV-2169 Salaspils, Latvia; 2Department of Aquatic Sciences and Assessment, Swedish University of Agricultural Sciences (SLU), SE-75007 Uppsala, Sweden; 3Institute of Ecology and Earth Sciences, University of Tartu, Vanemuise 46, 51003 Tartu, Estonia; 4Faculty of Geography and Earth Sciences, University of Latvia, Jelgavas Str. 1, LV-1004 Rigi, Latvia; 5State Key Laboratory of Environmental Geochemistry, Institute of Geochemistry, Chinese Academy of Sciences, Guiyang 550081, China

**Keywords:** Hg, freshwater sediment, water catchment area, sediment microbiome, bacterial community, archaeal community, amplicon-based next-generation sequencing

## Abstract

Inorganic mercury (Hg) can be methylated to the highly toxic and bioavailable methylmercury (MeHg) by microorganisms in anaerobic environments. The Hg methylation rate may be affected by forest management activities, which can influence the catchment soils, water, and sediments. Here, we investigate the influence of forest management in the form of ditch cleaning and beaver dam removal, as well as the seasonal variations, on sediment chemistry and microbiota. The relationships between MeHg concentrations in sediment samples and archaeal and bacterial communities assessed by 16S rRNA gene amplicon sequencing were investigated to determine the microbial conditions that facilitated the formation of MeHg. Concentrations of MeHg were highest in undisturbed catchments compared to disturbed or slightly disturbed sites. The undisturbed sites also had the highest microbial diversity, which may have facilitated the formation of MeHg. Low MeHg concentrations and microbial diversity were observed in disturbed sites, which may be due to the removal of organic sediment layers during ditch cleaning and beaver dam removal, resulting in more homogenous, mineral-rich environments with less microbial activity. MeHg concentrations were higher in summer and autumn compared to winter and spring, but the temporal variation in the composition and diversity of the microbial community was less than the spatial variation between sites. Beta diversity was more affected by the environment than alpha diversity. The MeHg concentrations in the sediment were positively correlated to several taxa, including *Cyanobacteria*, *Proteobacteria*, *Desulfobacterota*, *Chloroflexi*, and *Bacteroidota*, which could represent either Hg-methylating microbes or the growth substrates of Hg-methylating microbes.

## 1. Introduction

Mercury (Hg), and its organometallic form, monomethylmercury (MeHg), are highly toxic, and the latter can be biomagnified along aquatic food webs, resulting in contaminant levels in aquatic food resources that are unsafe for regular human consumption [1,2,3,4,5,6,7,8,9,10,11,12]. In large parts of the northern hemisphere, including the hemiboreal zone, Hg and MeHg concentrations in both freshwater and saltwater fish tissues often exceed the European Union threshold of 0.02 mg Hg kg^−1^ wet weight [13,14,15,16,17], especially in the Baltic Sea where Hg levels have been historically high [18]. MeHg is of a special concern, due to its high mobility and accumulation capacity in the higher levels of aquatic food webs [1,2].

The deposition of anthropogenic Hg has resulted in elevated Hg concentrations in soils and waters [12,19,20]. As the functional groups present within soil organic matter (OM) have a particularly high affinity for Hg [21,22], the retention of Hg in forest soils has increased [23]. Thus, OM content in soil can significantly affect the solubility, mobility, and toxicity of Hg in soils [21], and soil OM is an important vector of Hg and MeHg transport from terrestrial catchments to the surrounding surface waters, resulting in its accumulation in sediments that support downstream aquatic ecosystems [24,25,26].

Hg methylation is a microbial process, primarily carried out by dissimilatory sulfate-reducing bacteria (SRB), including species from the genera *Desulfovibrio*, *Desulfotomaculum*, and *Desulfobulbus* [27]; dissimilatory iron-reducing bacteria (FeRB), including *Geobacter sulfurreducens* [28]; by methanogens, such as *Methanomethylovorans hollandica* and *Methanolobus tindarius* [29,30]; and by fermenters [7]. A two-gene cluster, *hgcA* (encoding a putative corrinoid protein) and *hgcB* (encoding a 2[4Fe-4S] ferredoxin), is required for *Desulfovibrio desulfuricans* ND132 and *Geobacter sulfurreducents* to methylate Hg [31,32]. Methanogens, FeRB, and SRB are all facultative anaerobes [7].

Environmental and physiochemical factors, such as temperature, pH, the presence of electron acceptors (such as SO_4_^2−^) and electron donors (such as OM), the pool of available Hg, and redox conditions, can influence the abundance and activity of Hg-methylating microorganisms, which consequently affects the rate of MeHg formation [7,14,25,33,34]. In addition, the composition of the microbial communities will also influence the Hg methylation rate, not only in terms of the abundance of *hgcAB+* microorganisms that can carry out Hg methylation but also by acting as source substrates for these Hg-methylating microbes, which consequently modulate the activity of MeHg demethylation [7].

Potential sources of MeHg in aquatic systems include wetlands with OM-rich soils [35,36,37,38], reservoirs [39], and forested areas that have been flooded by beavers [14,40,41,42]. Forest operations can also catalyse the elevated formation and mobilization of Hg to surface waters [15,43,44,45]. However, many previous studies focused on the effects of forest harvesting operations and how these processes caused soils to become wetter. Here, we instead investigate the influence of forest management activities aimed at increasing water drainage in soils by cleaning ditches or removing beaver dams.

As the interactions between microbiology and chemistry are key to predicting Hg methylation rates [7,25,46,47,48,49], we specifically seek to investigate the microbial communities and the chemical composition of sediments in water courses. Sediments are considered to be one of the most representative environmental indicators of contamination due to their capacity to adsorb and accumulate Hg and MeHg in aquatic systems, even when they present at low concentrations in surface waters [50,51]. The aim of this study is to (1) test if forest management operations involving ditch cleaning and beaver dam removal influence the microbial communities and total Hg (THg) and MeHg concentrations in the sediments of water courses, and (2) evaluate how sediment THg and MeHg concentrations and microbial communities vary seasonally in these forest-managed catchments.

## 2. Materials and Methods

### 2.1. Sampling Sites

This study was conducted in the Kalsnava Forest district in eastern Latvia, comprising drained forests with organic-rich soil dominated by Scots pine (*Pinus sylvestris* L.), Norway spruce (*Picea abies* L. Karst.), and birch (*Betula* spp.) (accounting for 40.0%, 32.5%, and 16.1% of the area of studied catchments, respectively; Figure 1). In the study area, for the period 2018−2019, the mean annual precipitation was 636 mm, and the mean annual air temperature was 6.6 °C. The minimum mean monthly temperature recorded was −8.7 °C (February 2018), and the maximum mean monthly temperature recorded was 19.2 °C (July 2018). A dense drainage ditch network was initially established in the study area in the 1960s, with additional drainage works carried out in the 1980s. The bedrock is composed of sand of varying grain size, overlain by peat. The peat layer has a thickness greater than 20 cm over an area that accounts for 46.0% of the studied catchments (a total area of 1601.5 ha). The main forest management system is uniform regeneration felling; the proportion of clear-felled compartments currently varies between 0% to 6.2% of the catchment area. Some sections of the study area (including sampling site 3) are located in a protected area, the floodplain mire of the river Veseta. A total of eight sampling sites were selected from watercourses (including drainage ditches) across the study area, representing varying conditions and degrees of management-induced disturbance due to ditch cleaning and beaver dam removal (Figure 1, Table 1 and Appendix A). Both of these anthropogenic activities aim to increase the drainage of water at these sites, but they also cause disturbances in the ditches and streams.

### 2.2. Sampling and Processing of Sediments

Samples from the upper sediment layer (0–5 cm) were collected across the four seasons: autumn (21 November 2018), winter (14 February 2019), spring (21 May 2019), and summer (1 August 2019). Samples were collected using a submersible stainless steel sediment scoop that traps approximately 1.5 L of a sample as the instrument penetrates the sediment at the bottom of a water body. Excess water was released through a stainless-steel mesh mounted on the scoop using valves as the instrument was lifted out of the water. The equipment was washed with deionized water and sterilized with ethanol after each sampling attempt. The sediment sample was homogenised and split into two subsamples: one subsample was reserved for MeHg and DNA extraction, and the second subsample was used to determine the THg as well as other general chemical characteristics. Sediment subsamples intended for MeHg and microbiological analysis (200 mL) were immediately placed in sterile containers, frozen in liquid nitrogen, and transported on dry ice to the laboratory where they were stored at −80 °C until further analysis. Samples were freeze-dried for 72 h at −55 °C (STERIS GmbH, LYOVAC GT2-E) and sieved using a 1 mm stainless steel mesh. The sediment subsamples used for THg calculations and general chemical analysis (500 mL) were transported on dry ice to the LVS EN ISO 17025:2018 accredited laboratory at the Latvian State Forest Research Institute “Silava” (LSFRI Silava) and were prepared for analyses according to the LVS ISO 11464:2005 standard. Water samples were also collected at each sampling site for Hg, MeHg, and general chemistry analyses; these results were summarized in [52].

### 2.3. DNA Extraction, PCR Amplification and Sequencing

Total genomic DNA was extracted from 50 mg of the freeze-dried and sieved sediments using the DNeasy PowerSoil kit (Qiagen, Germany), with slight adjustments to the manufacturer’s instructions—after the first 5 min of vortexing (Step 4), samples were incubated in a water bath at 60 °C for 10 min, followed by an additional 10 min of vortexing. The quality and quantity of the extracted DNA were measured with a NanoDrop 8000 spectrophotometer (Thermo Scientific, Waltham, MA, USA).

PCR was performed with a set of bacterial and archaeal 16S (V4-V5) primers 515f [53] and 926r [54]. Each primer pair was tagged with a different molecular identifier. PCR reactions were prepared in a total volume of 25 µL, comprising 1 µL of template DNA (20 ng), 5 µL of Hot Fire Blend Mastermix (Solis Biodyne, Tartu, Estonia), and 0.5 µL of each forward and reverse primer (10 µM). The PCR conditions were as follows: initial denaturation of 95 °C for 15 min, followed by 26 cycles of denaturation at 95 °C for 30 s, annealing at 55 °C for 30 s, and elongation at 72 °C for 1 min, followed by at 72 °C for 10 min. Each sample was amplified in triplicate and pooled before purification using a FavorPrep PCR purification mini kit (Favogen, Taiwan). DNA from a blank extraction, a non-template PCR, and a microbial community DNA standard (D6305, ZymoResearch, Irvine, CA, USA) were used as controls. Final library preparation and sequencing were conducted in a core facility at the Institute of Genomics (Tartu, Estonia) using paired-end sequencing in an Illumina Miseq sequencer using 2 × 250 bp V2 chemistry.

### 2.4. Sequencing Data Analysis

The raw dataset consisted of 6,562,063 paired-end reads. Sequences were oriented in SEED2 [55], sorted in PipeCraft 1.0 [56], and 4,067,061 sequences were demultiplexed in QIIME 2 2020.2 [57]. DADA2 [58] was used for sequence filtering, trimming, and denoising; representative sequences were subsequently used to generate phylogenetic trees. The taxonomy of amplicon sequence variants (ASVs) was assigned using the high-quality ribosomal RNA database, SILVA (138, released in December 2019) [59], which was trained on the 16S gene regions V4-V5 using a Naive Bayes classifier. The samples had between 22,000 to 81,000 high-quality sequences (with a median of 52,192), representing a total of 17,385 ASVs. Then, 16S rDNA nucleotide sequences were deposited in the Sequence Read Archive (SRA) with accession numbers SAMN18879572–SAMN18879603 under BioProject ID PRJNA725443.

### 2.5. Chemical Analysis

The THg content of the sediment samples was determined at the LSFRI Silava using thermal decomposition, amalgamation, and atomic absorption spectrophotometry (Milestone DMA—80 AC-N) according to US EPA 7473, with a limit of detection of 0.001 mg kg^−1^. The MeHg concentration of the sediment samples was determined at the Institute of Food Safety, Animal Health and Environment “BIOR”, Latvia, using gas chromatography-inductively combined plasma-mass spectrometry (GC-ICP-MS; GC, Thermo Scientific Trace 1300; ICP-MS, Thermo Scientific iCAP RQ) according to BIOR-T-012-199-2019/1. The limit of detection was 0.04 ng g^−1^. %MeHg reflects a proportion of THg in the form of MeHg and is used as an indicator of the degree of Hg methylation.

General chemical characteristics were determined at LSFRI Silava. pH (CaCl_2_) was analysed according to LVS ISO 10390:2002 L/NAC:2005 L. Total carbon and organic carbon (TC and OC, g kg^−1^), total nitrogen (TN, g kg^−1^), and total sulphur contents (TS, mg kg^−1^) were determined using an elementary analysis method according to LVS ISO 10694:2006, LVS ISO 13878:1998, and ISO 15178:2000, respectively. The carbonate concentration, i.e., the inorganic carbon content (IC, g kg^−1^), was determined using volumetric methods according to LVS ISO 10693:2914. The total phosphorus content (TP) was determined using spectrophotometric methods according to LVS 298:2002 and LVS EN 14672:2006. HNO_3_-extractable potassium (K), calcium (Ca), and magnesium (Mg) concentrations were determined using inductively coupled plasma-optical emission spectrometry (ICP-OES). The electrical conductivity (EC, µS cm^−1^) of the sediment samples was determined according to LVS ISO 11265:1994. Content of chemical elements, including THg and MeHg in sediment samples was expressed per mass of dry samples.

MeHg data values below the detection limit (<0.04 µg kg^−1^, *n* = 4) were replaced by randomly generated data with values between 0 and 0.04 µg kg^−1^ such that the dataset followed a normal distribution (*µ* = 0.02, *σ* = 0.01) [60]. This was performed in order to not constrain MeHg and Hg data below the detection limit to one value—zero. The samples were categorised in terms of whether they had high or low Hg and MeHg concentrations as follows—low Hg: 3–33.5, high Hg: 33.6–187, and low MeHg: 0.011–1.315, high MeHg: 1.316–53.100, with each division selected based on median values. In this case, the labels “high” and “low” represent division relative to the range of the collected data and not in relation to other published estimates.

### 2.6. Bacterial and Archaeal Community and Statistical Analysis

The metadata and output files from QIIME 2 were combined for downstream analysis, performed in R (v4.0.3) [61]. Sequences that were not classified to the phylum level were removed in subsequent analytical steps. Any sample contamination was identified and removed using the R package decontam (v1.14.0) using the “prevalence” method at a threshold of 0.1 [62]. This step resulted in the removal of two ASVs. Following this, the control samples were removed, and the dataset was filtered for ASVs with a read count of more than ten reads to reduce the influence of rare ASVs. Finally, the samples were rarefied to a common sequence count of 22,000 sequences per sample, using the R package *vegan* (v2.5-7) [63].

Diversity metrics were calculated using the rarefied dataset. Alpha diversity was characterized using the observed species as well as the Shannon, Inverse Simpson, and Faith’s phylogenetic diversity indices using R packages phyloseq (v1.34.0) [64] and *picante* (v1.8.2) [65]. The environmental effects on Alpha diversity were tested using linear mixed models from the lme4 package (v1.1-30) [66] with a randomized level of disturbance.

Beta diversity was calculated based on the Bray–Curtis dissimilarity matrix and was visualized by the ordinate non-metric multi-dimensional scaling (NMDS) method (*k* = 3) using the vegan and phyloseq packages. We used the “envfit” function (from the vegan package) to visualize the potential relationship between soil properties and community structures by fitting the significantly correlated vectors (*p* < 0.05) of soil properties onto the coordinate space described by the NMDS plot. We used distance-based redundancy analyses (dbRDA) on a Bray–Curtis distance matrix of the environmental parameters to better visualize the possible relationship between the structure of the microbial community, its most representative class of bacteria or archaea, and environmental parameters.

The environmental effects on Beta diversity and the differences in community composition were tested using a permutational multivariate analysis of variance (PERMANOVA) with 9,999 permutations using the “adonis2” function available in the *vegan* package. We tested the average within-group dispersion of all groups using the “permutest” function available in the vegan package. Correlations between environmental variables and the community resemblance matrix (Bray–Curtis distance on a square root transformed ASV matrix) were assessed using the BEST routine (Bioenv method, Spearman rank correlation, 999 permutations) in PRIMER v7 [67]. Variables were transformed by square root (THg, TC, TN, TS, TP, K, Mg, Ca, EC, and %MeHg) or log (MeHg) functions and were normalized prior to the use of the BioEnv method.

Indicator analysis was performed on the ASV table using the labdsv (v2.0-1) package [68] to identify a list of taxa associated with each level of disturbance and methylmercury concentration. The following parameters were used for the indicator value analysis: 20,000 permutations, *p* value < 0.05 (false discovery rate adjusted), and indicator value > 0.5. We also performed a differential abundance analysis on the two contrasting groups (level of disturbance: major vs. undisturbed) using the DESeq2 (v1.34.0) package [69]. The results are reported in terms of the logarithmic (log2) fold change value.

The Pearson correlation coefficients between the environmental variables and taxa were visualized using a differential abundance test as well as a random forest analysis with subsequent correlation analysis, implemented using the microeco (v.0.11.0) package [70].

To account for the correlation between repeated measurements, linear mixed models were used to evaluate the impact of disturbances (levels of management-induced disturbance) on the physicochemical parameters of sediments. The dependence of the data within a catchment was addressed by including the “sampling site” as a random categorical factor.

The chemical characteristics of the sediment samples (X) were used to explain the variance in the annual mean THg and MeHg concentrations as well as %MeHg in sediments (Y) using a partial least squares (PLS) regression, implemented in *mdatools* [71,72].

## 3. Results

### 3.1. Sedimentary Hg Concentrations in Disturbed and Undisturbed Watercourses

The THg concentrations (Figure 2a) were significantly higher in the undisturbed sampling sites (Sites 1–3) compared to the disturbed sampling sites (Site 4–8; *p* < 0.001). The MeHg concentrations (Figure 2b) were also significantly higher in the undisturbed sampling sites compared to the disturbed sampling sites (*p* = 0.037). It appears that management-induced disturbance has no effect on the ratio between MeHg and organic C (Figure 2c) and the proportion MeHg compared to total Hg (Figure 2d).

PCA analysis on sediment chemistry data (Table 2) reveals the presence of two primary clusters (Figure 3). Samples from the disturbed sites (both major and minor disturbances) were clustered closely together, while the samples from undisturbed sites formed a more dispersed cluster. Loadings from variables primarily affected the first component of the axis.

The THg concentration of the sediment samples was positively correlated with the OC concentration (r = 0.88), TN concentration (r = 0.93), TS concentration (r = 0.88), TP concentration (r = 0.80), Mg concentration (r = 0.57), Ca concentration (r = 0.88), and EC (r = 0.72). The MeHg concentration of the sediment samples was positively correlated with the OC concentration (r = 0.62, *p* < 0.001), TS concentration (r = 0.69, *p* < 0.001), and Ca concentration (r = 0.53, *p* = 0.002) (Appendix A). %MeHg was either not correlated or only weakly correlated (r < |0.50|) with the general physicochemical characteristics of the sediments. Some physicochemical parameters also differed significantly between the undisturbed and disturbed sites, with the undisturbed sites exhibiting higher concentrations of OC, TN, TS, and Ca compared to disturbed sites (Table 2). The PLS analyses that attempted to explain the spatial variation of annual mean THg concentration in sediments (Appendix A) resulted in a strong model, with one significant component exhibiting excellent goodness of fit (R^2^ = 0.95) and goodness of prediction (Q^2^ = 0.87; full cross-validation). The variables that best explained the variation in THg concentrations (VIP > 1) were the TN, TS, and Ca concentrations in sediments as well as the proportion of pine-dominated forest compared to the total forested area. However, sites with a higher proportion of pine-dominated forest from the total forest area in each catchment were predominantly those without previous management-induced disturbance. Thus, management-induced disturbance, not tree species composition, in catchment may actually be the primary influencing factor. In contrast, the PLS analyses conducted on the mean annual MeHg and %MeHg resulted in weak models. These weak models could be due to a lack of linear relationships between MeHg and %MeHg and potentially relevant descriptors (the chemical characteristics of sediments as well as catchment characteristics).

### 3.2. Microbial Composition and Diversity in Sediments of Disturbed and Undisturbed Water Courses

#### 3.2.1. Bacterial and Archaeal Composition

The sequence dataset (*n* = 32) contained 1,686,738 reads and 17,420 ASVs. After taxa inspection, filtration, and rarefaction, 12,077 ASVs were left in the dataset (Appendix A). Following, 97.4% of sequences were assigned to phyla, 96% to class, 95% to order, 94% to family, 90% to genus, and 62% to species level. The average bacterial ASV richness (observed species) ranged from 689 to 2137 units per sample. From among 66 phyla, the most abundant phylum across the sampling sites was *Bacteroidota* (26% of all sequences), followed by *Proteobacteria*, *Cyanobacteria*, *Acidobacteriota*, *Chloroflexi, Verrucomicrobiota*, and *Desulfobacterota* (representing 25%, 10%, 8%, 8%, 7% and 5% of all sequences, respectively). There were 166 bacterial and archaeal classes, of which the most abundant were *Bacteroidia* (26%), *Gammaproteobacteria* (22%), *Cyanobacteria* (13%), *Alphaproteobacteria* (9%), and *Verrucomicrobiae* (8%).

In total, 4.6% of all sequences in the rarefied dataset were archaea, belonging to, in order of abundance, *Crenarchaeota*, *Nanoarchaeota*, *Thermoplasmatota*, *Halobacterota*, and *Euryarchaeota* (Appendix A).

The largest difference in microbial composition across all levels of disturbances could be attributed to *Cyanobacteria*, where 28% of all sequences from sites that had been subject to major disturbances were *Cyanobacteria*, while *Cyanobacteria* only accounted for 2% and 4% of all sequences from undisturbed or minimally disturbed sites, respectively (Figure 4).

Correlations between the environmental variables and the general taxa were similar for THg, TN, Ca, TS, OC, and TC (Appendix A). The concentration of MeHg in sediments was positively correlated with the abundance of sequences from the classes *Bacteroidota, Spirochaetota, Nanoarchaeia, Omnitrophia*, and *Kryptonia*.

#### 3.2.2. Alpha and Beta Diversity

There were significant differences between the Shannon and inverse Simpson indices across the various disturbance levels (Figure 5a,b). Post hoc comparisons using the Tukey HSD test indicated that the mean Shannon and inverse Simpson values for the undisturbed sites (M = 6.78, SD = 0.28 and M = 513.24, SD = 172.62, respectively) and the sites with minor disturbances (M = 6.61, SD = 0.40 and M = 509.58, SD = 218.71, respectively) was significantly higher than their values in the sites with major disturbances (M = 6.09, SD = 0.38 and M = 149.99, SD = 88.85, respectively). Differences between observed species (Figure 5c) and Faith’s phylogenetic diversity (PD) (Figure 5d) were not significant across the various disturbance levels.

Sites with higher concentrations of Hg had significantly higher Shannon and inverse Simpson indices (Appendix A), but species richness (Appendix A) and PD (Appendix A) were similar for groups with high and low concentrations of Hg. Sites with a high concentration of MeHg had significantly higher Shannon (Appendix A) and inverse Simpson values (Appendix A), as well as higher PD values (Appendix A) and greater richness indices (Appendix A).

Although strong correlations were observed between Alpha diversity metrics and most of the environmental variables, only the pH values were significantly correlated with the Simpson (r(30) = 0.48, *p* = 0.005483) and Shannon (r(30) = 0.47, *p* = 0.006827) indices (Appendix A). Furthermore, linear mixed models indicated that the relationship between the environmental variables (THg + MeHg + Ca + TC + TN + TS + Ca + EVS) and the levels of disturbance did not have a significant effect on the Shannon and inverse Simpson indices.

The results of NMDS are summarised in Figure 6. Three dimensions were used for ordination, reaching a stress value of 0.10. The PERMANOVA analysis revealed that, of the variation observed in community composition (Bray–Curtis dissimilarity matrix), 25% was due to the level of disturbance (R^2^ = 0.24999, F = 4.8331, *p* < 0.001), 8% was due to the level of MeHg (R^2^ = 0.08065, F = 2.6317, *p* < 0.001), and 10% was due to the level of Hg (R^2^ = 0.09802, F = 3.2602, *p* < 0.002). Disturbed sites were less spatially dispersed on the ordination plot compared to undisturbed sites. However, a permutation-based test of the multivariate homogeneity of dispersed groups indicated that the between-group variation is greater than the within-group variation for all of the factors previously mentioned, with *p* = 0.095, 0.102, 0.567 and 0.291 for the level of disturbance, the level of MeHg, the level of Hg, and the change in seasons, respectively.

Most of the environmental variables fitted on the ordination plot with “envfit” (vegan package) exhibited an association with the undisturbed sites on the first axis derived by NMDS (Figure 6). We used PERMANOVA to test the effects of continuous environmental variables on the composition of the microbial community within a single model. We found significant evidence that the composition of the microbial community was influenced by a variety of physicochemical variables in soil, such as MeHg (R^2^ = 0.07452, F = 3.1196, *p* < 0.001), THg (R^2^ = 0.09526, F = 3.9879, *p* < 0.001), TC (R^2^ = 0.06196, F = 2.5939, *p* = 0.002), pH (R^2^ = 0.05265, F = 2.2041, *p* = 0.006), TP (R^2^ = 0.04274, F = 1.7893, *p* = 0.012), and K concentrations (R^2^ = 0.03917, F = 1.6397, *p* = 0.034).

Similarly, the BEST routine (BioEnv) revealed the set of environmental variables that were the most well-correlated with the microbial resemblance matrix. The highest correlation (Spearman’s rank correlation coefficient of 0.535) involved five variables: TS, TC, and Mg (square root-transformed), MeHg (log-transformed), and pH. The single most well-correlated variable was TC, with a correlation of 0.512, while Hg had a correlation coefficient of 0.431. Sample statistic (Rho) for the BEST routine was 0.535, *p* = 0.001. We also conducted a redundancy analysis (RDA) test on this set of variables. It was revealed that the five explanatory variables explained 23% of the total variation in the composition of the microbial community. However, its predictive relationship with the composition of the microbial community was debatable, as the Monte Carlo permutation (999 permutations) test for the model was not significant (F(4, 27) = 2.0681, *p* = 0.051) (Figure 7).

#### 3.2.3. Differential Abundance Testing and Indicator Analysis

We observed an increase in reads of photosynthetic bacteria (*Cyanobacteria*) in disturbed sites and sites with lower levels of Hg and MeHg. A microbial abundance heatmap of samples grouped according to high and low concentrations of MeHg reveals the existence of a distinct cluster, primarily consisting of *Cyanobacteria*, *Crenothrix*, and *Terrimonas* at Sites 5 and 6 (disturbed), which exhibited a low concentration of MeHg (Appendix A). Undisturbed sites characterized by high levels of MeHg lack this cluster (Appendix A). These observations were also supported by DESeq2 analysis. Many genera of *Cyanobacteria* and *Proteobacteria* were more abundant in sites subjected to major disturbances than in undisturbed sites (Appendix A). Sites with a higher concentration of MeHg had a higher abundance of *Myxococcota*, *Planctomycetota*, *Nanoarchaeota*, and *Bacteroidota* and a relatively lower abundance of *Cyanobacteria* and *Proteobacteria* (Appendix A).

Indicator analysis identified numerous taxa that could be attributed to various factor levels (i.e., the level of disturbance, MeHg, or Hg; Appendix A, as well as the summary in Appendix A). In total, 132, 330, and 212 taxa were identified as indicator species for sites with minor, major, and no disturbances, respectively. Furthermore, 113 and 43 ASVs were found to be indicator species for high and low concentrations of MeHg, respectively, and 110 and 112 ASVs were found to be indicator species for high and low concentrations of Hg, respectively. However, after adjusting the *p* values to account for multiple testing, no significant taxa could be assigned to different levels of Hg and MeHg concentrations. The distribution of indicator species by phyla is summarized in Appendix A.

### 3.3. Seasonal Variations in Sedimentary MeHg and Microbial Community Composition

The highest mean MeHg concentrations in the sediment samples were observed in summer (11 ± 5 µg kg^−1^) and autumn (11 ± 6 µg kg^−1^). The MeHg concentrations in summer and autumn were also significantly higher than the concentrations recorded in winter and spring (*p* = 0.004).

Similarly, the %MeHg in the sediment samples was higher in summer and autumn than in winter and spring (*p* < 0.001). Furthermore, the highest mean MeHg/OC ratio was observed in the summer and autumn seasons (0.061 ± 0.013 and 0.058 ± 0.013 µg g^−1^, respectively).

Although the Alpha diversity and the InvSimpson values differed across the disturbance levels, seasonality did not affect these measures (one-way ANOVA; Appendix A). There was also no distinct seasonal pattern identified after NMDS analysis was conducted based on the Bray–Curtis dissimilarity matrix; these findings were supported by a PERMANOVA test. In general, the variation in microbial communities and the MeHg concentrations in the sediments in the context of seasonality was lower than the variation between sites and the level of disturbance.

## 4. Discussion

### 4.1. The Effect of Disturbances on the MeHg and THg Concentrations in the Sediment

The THg concentration in the sediment samples was significantly higher in the undisturbed sites compared to the disturbed sites. In one of the undisturbed sites (Site 1), the %MeHg value had a seasonal maximum of up to 34% and 36% during the summer and autumn seasons, respectively. Comparatively high %MeHg values in the undisturbed sites indicated relatively intense Hg methylation rates, resulting in high MeHg concentrations at these sites. MeHg concentrations and %MeHg were also significantly higher in the undisturbed sites compared to the disturbed sites sampled. In addition, many sedimentary chemical characteristics, such as OC and TS concentrations, differed across disturbance levels. The high abundance of carbon and nutrients in undisturbed sites could favour Hg methylation by resulting in higher decomposition rates, which consume oxygen and lower the sediment’s redox status; in addition, some elements, such as carbon and sulphur, can act as electron donors and acceptors for Hg methylating microorganisms [23]. Furthermore, the MeHg concentrations were more closely associated with environmental conditions (e.g., temperature and redox potential) and the general chemical characteristics of the sediment than with THg concentrations, indicating that it is the surrounding environment, and not the THg concentration, that drives MeHg formation in these sediments. Several studies have highlighted the importance of the role of sulphate (SO_4_^2−^) and other organic sulphur species in stimulating MeHg formation in wetlands [33,73,74,75]. We also detected a correlation between the TS content and MeHg concentrations in sediments (r = 0.69), while the correlation between THg and MeHg concentrations was much weaker (r = 0.49).

Although previous studies have found that forest harvesting operations increased the formation of MeHg [15], this study found that the disturbed sites had lower MeHg formation rates compared to the undisturbed sites. Previous studies primarily focused on forest harvesting operations, and the lower transpiration rates following tree removal can result in wetter areas and higher groundwater levels [76]. In this study, we evaluated forest management activities that aimed to lower the groundwater levels by increasing the drainage in these sites. In addition, both ditch cleaning and beaver dam removal result in the mechanical disturbance of the ditch bed, removing the upper layers of organic- and nutrient-rich sediment. The lower carbon and nutrient concentrations recorded in the disturbed sites support this interpretation. The removal of these organic- and nutrient-rich sediments appears to lower the MeHg formation rate. This suggests that the MeHg formation rates in catchment soils may decrease if the groundwater levels are lowered.

### 4.2. The Effect of Disturbances on the Microbial Community in the Sediment

Our study revealed that the highest concentrations of THg and MeHg were found in sediments from natural, undisturbed sites. This suggests that human intervention may be causing the transport of Hg compounds, but it also indicates that carbon and nutrients make the sedimentary environment favourable for MeHg formation. The composition of the microbial community in the sediment differs between disturbed and undisturbed sites. The lower microbial diversity in the disturbed sites, as indicated by the alpha diversity metrics (Shannon and Simpson indices), may be due to the loss of the organic- and nutrient-rich sediments described in the previous section, which has created a more homogeneous environment for the microorganisms. In the undisturbed sites, the decomposition of organic matter and the consumption of oxygen may have created more heterogeneous redox environments, allowing for the creation of anaerobic microhabitats. Former studies have also identified that forest harvesting activities may create reducing microhabitats that are favourable for Hg methylation [43]. Fuhrmann et al. [77] also observed elevated MeHg formation rates after organic carbon was added to freshwater sediment, most likely affecting microbial communities by causing the ambient environmental conditions to be moderately or highly reduced. Not only did the microbial diversity differ between the disturbed and undisturbed sites, but there were also differences in the taxa that were favoured in each site. The disturbed sites may have possessed physiochemical conditions that were favourable to certain taxa, such as *Cyanobacteria*. Previous reports have shown that disturbed sites have higher concentrations of dissolved carbon and total nitrogen in the water than undisturbed sites [52].

Although alpha diversity cannot be solely used to characterize microbiomes [78] and can find associations with Hg methylation, Frossard et al. [79] conducted a similar study and observed higher microbial richness and Shannon indices in soils from sites associated with long-term, high levels of Hg contamination. They also found that these indices were associated with the individual characteristics of the sites and suggested that microbial communities might be able to preserve their functional diversity along the Hg gradient. Furthermore, there were no visible short-term effects when Hg was added to microcosms [79].

Local physicochemical characteristics can be influenced by microbial activity [25,80]; this was not assessed in this study. We evaluated the presence, rather than the activity, of gene sequences in this study. Nevertheless, we used three statistical methods to identify possible relationships between the community composition and environmental variables. The PERMANOVA analysis indicated that almost all variables had some small but significant effect on community structure; however, the BioEnv method identified a smaller set of variables with the highest correlation (Spearman correlation = 0.535) with a resemblance matrix consisting of only five variables—TS, TC, Mg, MeHg, and pH. The envfit function suggested that most environmental variables may be primarily associated with undisturbed sites.

Jones et al. [81] showed that Hg-methylating communities that are dominated by diverse anaerobic microorganisms that do not reduce sulfate can produce MeHg as effectively as communities that are dominated by sulfate-reducing populations. Thus, the range of possible Hg-methylating communities could be very diverse. Some studies have highlighted the possibility of a mutualistic relationship between several species that promote Hg(II) methylation [80,82]. To this date, most studies on MeHg have focused on a small number of taxonomic groups, such as DSRB, DIRB, and some methanogens; however, there are growing concerns that some crucial taxa may have been overlooked [7]. The *hgcAB* gene sequences represent a unique opportunity to identify taxa that can methylate Hg [83]. However, it is clear that the presence of *hgcAB* genes will not indicate the presence of symbiotic microorganisms or any microorganisms that provide growth substrates for the Hg methylators. Thus, using a variety of diversity indices would be an excellent complement to analyses of *hgcAB* genes when identifying areas of high Hg methylation rates; these methods were not applied in this study. Instead, we found some taxa and communities that were associated with the level of disturbances experienced by the sites. Differential abundance and indicator analysis also identified several bacterial and archaeal groups that were more abundant in high MeHg sediments (Appendix A). Most of the taxa that were found to be specific to the undisturbed sites and high MeHg concentrations have previously not been related to Hg methylation or MeHg demethylation, such as *Bacteroidota* (*Bacteroidales*), *Nitrospirota* (unspecified), *Cholroflexi, Desulfobacterota*, *Spirochaetota*, *Crenarchaeota* (*Bathyarchaeia*), *Myxococcota* (*Haliangium*), *Proteobacteria*, and *Planctomycetota* (unspecified). DESeq2 associated at least two genera from previously known phyla of Hg methylators, *Desulfobacterota* (*Geobacter*) and *Spirochaetota* (*Leptospira*), with sites that had the lowest concentrations of MeHg. However, many of the taxa associations were found to be only marginally significant after correction for multiple testing; thus, this indicator analysis should be treated with caution [84], especially since the ability to methylate Hg can be strain-specific even for closely related organisms [85].

Low MeHg concentrations were associated with a low abundance of *Desulfobacterota* and *Chloroflexi* and a high abundance of *Cyanobacteria*. One possible connection between *Cyanobacteria* abundances and the low levels of MeHg was suggested by Lei et al. [86]: a dynamic set of processes might be happening in the eutrophic water where algal or similar organic material form. For example, *Cyanobacteria* could have a biodiluting effect on MeHg. It has been reported that *Cyanobacteria* can remove low levels of Hg species from aqueous environments, although up to 3% of the Hg removed was methylated to MeHg [82]. However, it is necessary to study associations between *Cyanobacteria* and other microbes in polluted environments. Evidence linking *Cyanobacteria* to MeHg production is inconsistent. Many microorganisms carrying *hgcAB* genes might benefit from the polysaccharides found in cyanobacterial blooms [87]. The authors of [88] reported a net increase in MeHg production in periphyton when cyanobacteria were present based on experimental evidence.

It is quite unsurprising that there was no strong correlation between the concentration of MeHg and THg and the copy number of *hgcAB* in other studies; in addition, the concentration of MeHg in the environment should not be considered an indicator for the activity of methylation processes [89]. There are several reasons for this: (1) *hgcAB* strains are rare (<1%) and are not often linked to 16S rRNA databases (<300 species) [85]; (2) it is important to consider that net MeHg production also depends on the activity of MeHg demethylation [7,77,89]. Thus, the absence of a correlation between the *hgcA* gene and MeHg concentration is not surprising and highlights the complexity of predicting MeHg concentration dynamics in the natural environment. The biological mechanisms involved in the full bio-physico-chemical process of net MeHg production remain poorly understood and must be described more accurately in future studies.

Understanding the mechanisms behind MeHg formation dynamics will help predict its movement under changing climatic and anthropogenic conditions [90]. This study provides preliminary insight into modelling MeHg dynamics in sediments from small watercourses in drained forests with organic soils.

### 4.3. Temporal Variation in Sediment MeHg and Microbial Communities

The MeHg concentration in sediments varied less over time than between sites, although its concentrations were found to be significantly higher in summer and autumn than in winter and spring. No seasonal patterns were observed in the composition of the microbial communities. The disturbance level of the sites explained the greatest proportion of the variance observed in the composition of the bacterial and archaeal communities in the sediment samples. This highlights the resilience of microbial communities in the sediment and explains their stability throughout the year. However, higher MeHg formation rates were commonly observed in the sediments during the warmer seasons [91].

## 5. Conclusions

The management-induced disturbance of watercourses, primarily in the form of ditch cleaning, significantly affected the physiochemical conditions of the sediments, especially in terms of decreases in the amount of OM. Previous studies have shown that forest harvesting resulted in elevated MeHg formation. In the sediments of watercourses where ditch cleaning and beaver dam removal were conducted, the forest management disturbances instead lowered the formation of MeHg. The diversity of microbial communities was also higher in sites with high concentrations of MeHg, most likely because these organic-rich, undisturbed sediments encouraged the proliferation of bacteria living at the oxic-anoxic boundaries, including Hg-methylating species.

Based on DNA analyses, the microbial community profile was found to be stable over the year-long sampling process. The seasonal variations in the MeHg concentrations in sediments were found to be less than the variation between sites. In this study, the characteristics of each site, in particular, the disturbance level, were found to influence the MeHg formation rates and the diversity and composition of microbial communities compared to any seasonal variations.

Thus, in addition to impact of seasonality, ditch and stream cleaning in drained forests with organic-rich soils can promote conditions (both physicochemical parameters of sediments and composition of microbial communities) less favourable for Hg methylation than in undisturbed watercourses.

## Figures and Tables

**Figure 1 microorganisms-10-01981-f001:**
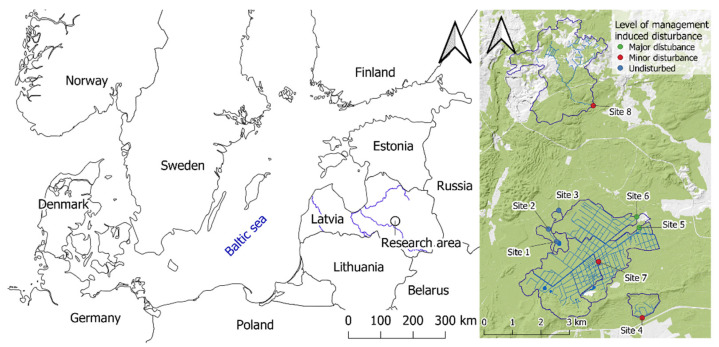
The research area, located in the experimental forests of the Kalsnava Forest district, eastern Latvia. Maps were made in QGIS Desktop (v3.16.3).

**Figure 2 microorganisms-10-01981-f002:**
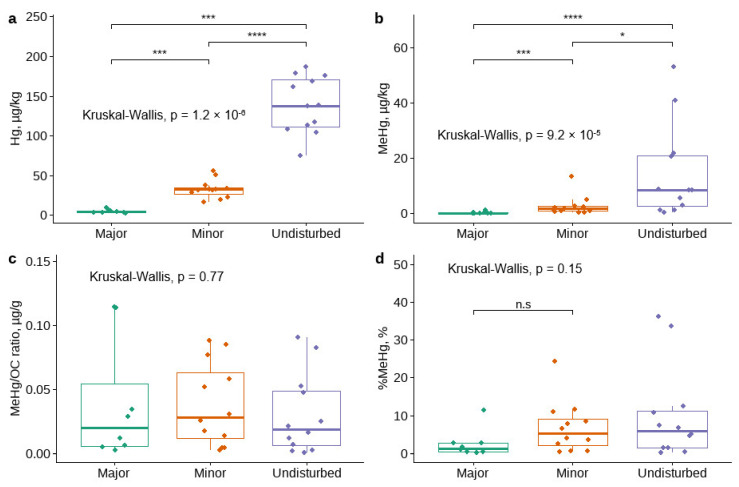
Variations in (**a**) MeHg and (**b**) THg concentrations, (**c**) MeHg/OC ratios, and (**d**) %MeHg in sediments as a function of the level of management-induced disturbance in watercourses. Asterisks above the bar plots indicate the statistically significant differences between the levels of disturbance according to Kruskal-Wallis test (not significant (n.s): *p* > 0.05; *: *p* ≤ 0.05; ***: *p* ≤ 0.001; ****: *p* ≤ 0.0001).

**Figure 3 microorganisms-10-01981-f003:**
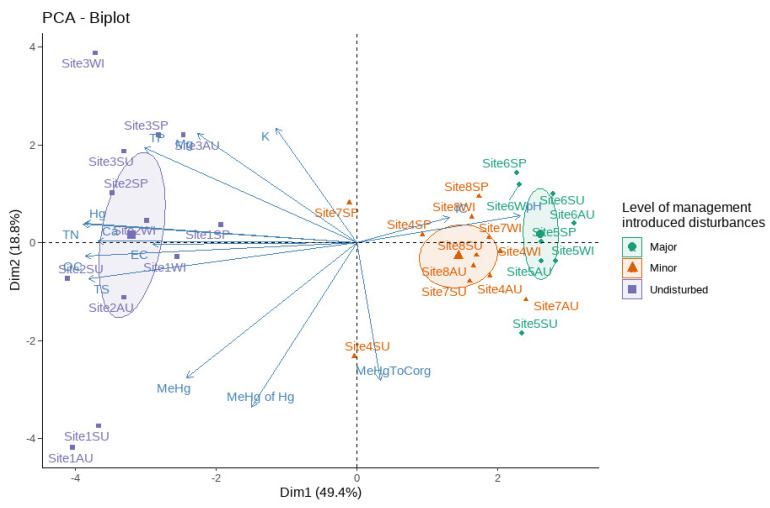
Physicochemical characteristics of the sampling sites. The ordination biplot of the PCA results shows that the sites are grouped by the differences in their physicochemical parameters.

**Figure 4 microorganisms-10-01981-f004:**
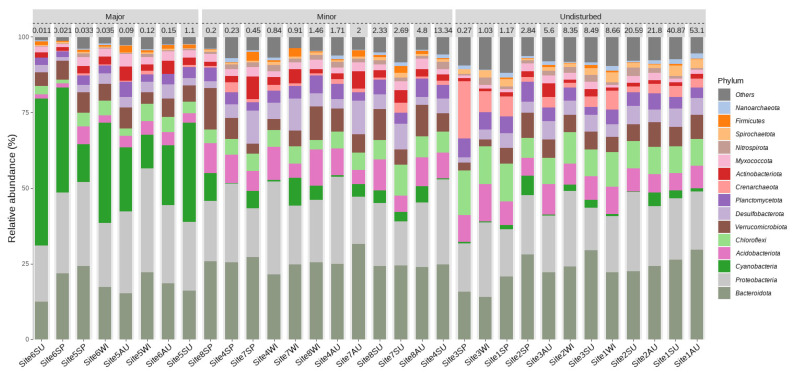
A bar plot describing the relative abundance of various phyla as a function of the level of disturbance. Samples with 15 the most abundant phyla among all the samples are grouped by the level of disturbance and are sorted from lowest to highest concentration (µg kg^−1^) of MeHg within each level.

**Figure 5 microorganisms-10-01981-f005:**
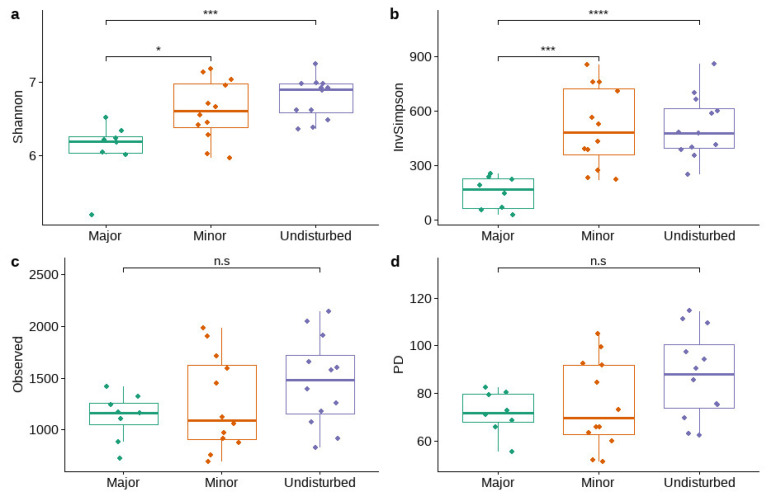
Alpha diversity metrics ((**a**)—Shannon index, (**b**)—Inverse Simpson index, (**c**)—observed species, (**d**)—Faith’s phylogenetic diversity) for sites of different disturbance levels. Asterisks above the bar plots indicate the statistically significant differences between the levels of disturbance based on one-way ANOVA (not significant (n.s): *p* > 0.05; *: *p* ≤ 0.05; ***: *p* ≤0.001; ****: *p* ≤0.0001).

**Figure 6 microorganisms-10-01981-f006:**
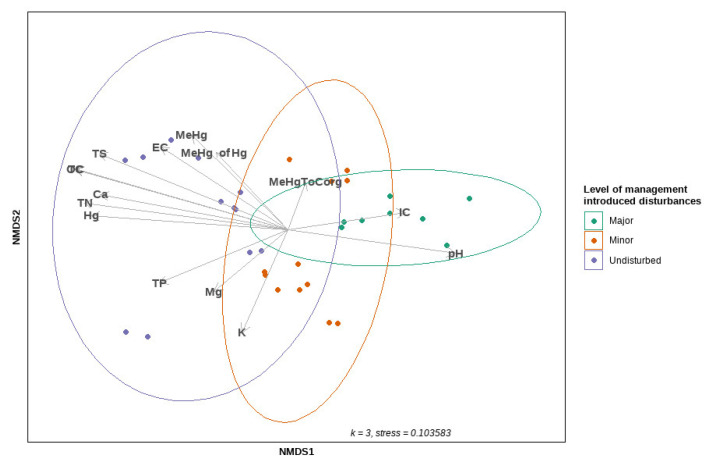
The ordination plot of non-metric MDS, using Bray–Curtis dissimilarity distances on the square root-transformed community data (abundance matrix); 95% confidence ellipses are fitted around centroids that represent the levels of management-induced disturbances, along with the environmental variables fitted on the ordination plot using the “envfit” function (vegan package).

**Figure 7 microorganisms-10-01981-f007:**
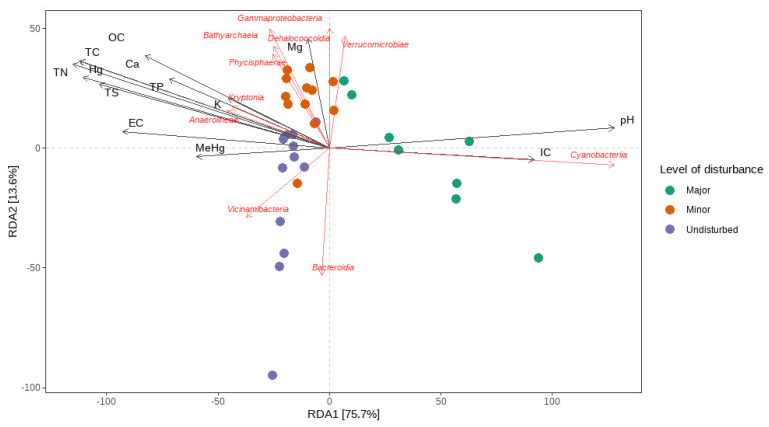
The ordination plot of the distance-based redundancy analysis (RDA). The ordination plot showcases the vectors representing the environmental variables as well as important taxonomical classes.

**Table 1 microorganisms-10-01981-t001:** Characterization of the sampling sites.

Level of Management-Induced Disturbance	Sampling Sites	Sampling Sites Coordinates (Latitude; Longitude)	Description
Undisturbed	Site 1	56.689° N; 25.815° E	Natural stream from a peatland lake to river, close to the outlet
	Site 2	56.693° N; 25.810° E	Natural stream from a peatland lake to river, 200 m downstream
	Site 3	56.699° N; 25.816° E	Natural stream with beaver site, nature reserve
Minor	Site 4	56.664° N; 25.863° E	Drainage ditch, undisturbed for several decades
	Site 7	56.682° N; 25.838° E	Drainage ditch, undisturbed for several decades
	Site 8	56.731° N; 25.839° E	Natural stream with slightly disturbed beaver site in recent years
Major	Site 5	56.693° N; 25.862° E	Drainage ditch with beaver dam removal in 2017
	Site 6	56.696° N; 25.861° E	Drainage ditch, ditch cleaning in 2016

**Table 2 microorganisms-10-01981-t002:** General chemical characteristics of the sediments compared to the level of management-induced disturbance in the watercourses. Different letters represent statistically significant differences between groups representing different levels of management-induced disturbance in the watercourses (*n* = 32).

Parameter, Unit	Value	Level of Management-Induced Disturbance of Watercourses		
		Undisturbed	Minor	Major
pH	mean ± S.E.	6.2 ± 0.1 ^a^	6.5 ± 0.1 ^a^	7.0 ± 0.2 ^a^
	range	5.8–6.7	5.7–7.1	6.3–7.6
EC, µS cm^−1^	mean ± S.E.	391 ± 49 ^a^	183 ± 48 ^ab^	54 ± 12 ^ac^
	range	135–665	30–601	23–122
OC, g kg^−1^	mean ± S.E.	448 ± 20 ^a^	66 ± 10 ^b^	4 ± 1 ^c^
	range	342–584	26–157	1–10
IC, g kg^−1^	mean ± S.E.	<0.1	<0.1	0.9 ± 0.5
	range	<0.1–<0.1	<0.1–<0.1	<0.1–4.0
TN, g kg^−1^	mean ± S.E.	22.3 ± 0.9 ^a^	4.2 ± 0.6 ^b^	0.4 ± 0.1 ^c^
	range	18.3–28.9	2.3–9.3	0.1–1.1
TS, mg kg^−1^	mean ± S.E.	6568 ± 486 ^a^	781 ± 174 ^b^	0.5 ± 0.5 ^b^
	range	4411–9644	196–2365	<0.1–4.3
TP, g kg^−1^	mean ± S.E.	0.79 ± 0.096 ^a^	0.39 ± 0.04 ^a^	0.31 ± 0.06 ^a^
	range	0.46–1.52	0.22–0.75	0.20–0.73
K, g kg^−1^	mean ± S.E.	1.16 ± 0.43 ^a^	0.58 ± 0.20 ^a^	0.33 ± 0.16 ^a^
	range	0.11–4.11	0.17–2.53	0.08–1.42
Mg, g kg^−1^	mean ± S.E.	2.83 ± 0.35 ^a^	1.03 ± 0.15 ^a^	1.58 ± 0.48 ^a^
	range	1.57–5.38	0.44–1.90	0.25–3.20
Ca, g kg^−1^	mean ± S.E.	30.9 ± 2.4 ^a^	6.4 ± 1.8 ^b^	4.9 ± 1.6 ^b^
	range	7.8–38.4	0.5–23.3	0.2–10.6

## Data Availability

Not applicable. 16S rDNA nucleotide sequences are openly available in the Sequence Read Archive (SRA) with accession numbers SAMN18879572–SAMN18879603 under BioProject ID PRJNA725443.

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
