# Peer review of "The Influence of the Degree of Forest Management on Methylmercury and the Composition of Microbial Communities in the Sediments of Boreal Drainage Ditches"

_microorganisms, 2022, doi:10.3390/microorganisms10101981_

Round 1

Reviewer 1 Report

Dear Authors,

Your manuscript ID: microorganisms-1920212, focus on important issue that is environment pollution by Hg in context of forest management and changes of microbial community. I have a few suggestion to this manuscript.

Check the order of figures, it should be accordance to order of citation in manuscript.

Check that the figures are cited correctly, according to the description in the text.

 Fig. 1. Line 108

Legend is unreadable. Remove one of the full stop at the end of the figure caption.

 Lines 133, 468, 478

Give details, the author's name in addition to the number of the cited publication in bracket. As it stands, the sentence feels like it has no ending.

 Line 166, paragraph “2.5. Chemical analysis”

TC, OC, IC, and other parameters were express per mass of dry samples? Add information.

 Line 219

Use capital letter for writing taxonomic unites.

 Line 328

Figures 5c and 5d are not mentioned in the manuscript.

 Line 334

Add a proper figures citation.

Line 525

Remove the year.

Reviewer 2 Report

The article is devoted to a comprehensive analysis of the relationship of various factors that can affect the content of methyl-mercury in the forest soils of eastern Latvia. The work is interesting and contains a number of new findings shedding light on the very complex relationships of physical, chemical factors and microbiological processes affecting MeHg formation dynamics. It seems to me that by its topic it is more suitable for publication in the journal Ecologies (MDPI) (ISSN 2673-4133). The manuscript is logically and clearly written, all the data are well illustrated and discussed. However, some phrases are poorly formulated. In addition, I have a few minor comments and questions, which are listed below.

Moreover, this can be considered a quibble, but I had difficulties in quickly understanding the drawings that the authors used unusual color designations that do not correspond to the commonly used. Usually red color means something that deviates greatly from the norm, and vice versa, the green color is something basic, and in this work green indicates the most disturbed areas, and red - undisturbed. In addition, for colorblind people, red and green are generally indistinguishable. Also on heat maps the most intense color usually indicates the maximum value of the parameter, and in this work, vice versa. Of course, it is the right of the authors to choose the colors for the designation, but in the interests of the reader, I would use more usual color schemes.

The Introduction does not clearly state why mercury methylation is so important to monitor.

L38-40: “Mercury (Hg), and its organometallic form, monomethylmercury (MeHg), are highly toxic and can be biomagnified along aquatic food webs…”- How can biological activity increase mercury content? This needs to be clarified.

In the Introduction or Discussion, you need to specify that the gene pair hgcAB is essential for microbial mercury methylation.

Please specify in the methods what %MeHg means. –Is it % of the total Hg content?

Mercury and MeHg content should be added to the Table 2.

The order of the panels in Figure 2 should be changed in accordance with the order of references to them in the text. In the text, indicate Figure 2a, 2b, etc.

L282-284: “The variables that best explained the variation in THg concentrations (VIP > 1) were the TN, TS, and Ca concentrations in sediments as well as the proportion of pine-dominated forest compared to the total forested area.” - Do you mean that there was more mercury where pine trees predominated in the forest? It can be seen from Table S1 that such forests were in all undisturbed ecosystems, in sites 1,2,3. Therefore, there may not actually be a connection of increased mercury content with such forests.

L 334 Fig S7a and S7b not figure7!!!

376 “However, their”, instead “However, its”? Or did I not understand the meaning of the sentence?

Tables S2-S4 are cropped and not fully presented. Therefore, I have a question: are their headings correct? Do you mean species or genera?

L461-463: “The lower microbial diversity in the disturbed sites, as indicated by the alpha diversity metrics (Shannon and Simpson indices), may be due to the loss of the organic- and nutrient-rich sediments described in the previous section, which has created a more homogeneous environment for the microorganisms.”- As far as I understand, when cleaning the ditches, part of the organic-rich sediment was removed from their walls and bottom. But in this way, not only the nutrients were removed, but also the bacteria themselves that live in them. But when the beaver dams were removed, this did not happen. Was there a difference in microbial communities due to different types of forest management works?

L536-550 It is a very interesting review of the published data, but it is difficult to grasp the connection with the results obtained in this study.

L551-561 Yours data show that mercury methylation most likely depends on the biological activity of the microbial community. To assess it, indicators such as the intensity of carbon dioxide emission, nitrogen fixation, ammonification, nitrification, denitrification, etc. are usually used. It makes sense to isolate and analyze RNA from the soil if you are going to check the activity of some specific genes, for example hgcAB.

L560 “This study account only for the nonactive genes (16S DNA); a seasonal pattern may have emerged if the active genes (RNA) were considered.” -  It is wrong to say this, since a significant part of copies of 16S RNA gene may be active. It's just that this method does not allow you to evaluate it.
